# The Standard Model of Rational Risky Decision-Making

Kazem Falahati 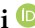

Glasgow School *for* Business and Society, Glasgow Caledonian University, 70 Cowcaddens Road,
Glasgow G4 0BA, UK; kfa@gcu.ac.uk; Tel.: +44-141-331-3708

**Abstract:** Expected utility theory (EUT) is currently the standard framework which formally defines rational decision-making under risky conditions. EUT uses a theoretical device called von Neumann–Morgenstern utility function, where concepts of function and random variable are employed in their pre-set-theoretic senses. Any von Neumann–Morgenstern utility function thus derived is claimed to transform a non-degenerate random variable into its certainty equivalent. However, there can be no certainty equivalent for a non-degenerate random variable by the set-theoretic definition of a random variable, whilst the continuity axiom of EUT implies the existence of such a certainty equivalent. This paper also demonstrates that rational behaviour under utility theory is incompatible with scarcity of resources, making behaviour consistent with EUT irrational and justifying persistent external inconsistencies of EUT. A brief description of a new paradigm which can resolve the problems of the standard paradigm is presented. These include resolutions of such anomalies as instant endowment effect, asymmetric valuation of gains and losses, intransitivity of preferences, profit puzzle as well as the St. Petersburg paradox.

**Keywords:** decision-making; rationality; risk; expected utility; behavioural puzzles

**MSC Codes:** 62Cxx; 90B50; 91A26; 91B02; 91B06; 91B16; 91B30

**JEL Codes:** C00; D01; D81

## 1. Introduction

It is generally accepted that science is *provisional* by nature. To confirm, improve or reject it, it must be open to impartial and rigorous re-examination. It is this openness to challenge which distinguishes science from dogma and mythology. Without such scrutiny, resolutions of existing scientific anomalies cannot emerge, hence the need for them. However, any rejection of the extant views as a result of such a re-examination is a bitter pill to swallow for unsuspecting holders of those views, and hence their understandable surprise and resistance to a change of views, a resistance which must be overcome for science to progress. In this context, this paper aims to re-examine expected utility theory (EUT) by first setting out the claims of EUT and the problems that they generate, and later by identifying the *generic* cause of these problems and finally indicating how they can be resolved. This article peruses the issues that EUT raises, with the care and respect that they deserve, i.e., rigorously and thoroughly from the perspectives of the disciplines of mathematics/statistics *and* economics/finance, hence the length of it.

Assuming the axioms of EUT hold for all gambles, the utility of each gamble from the set of all gambles available to a *rational* decision-maker is claimed to be the statistical expectation of the utility of its outcomes (e.g., Jehle and Reny 2011, pp. 97–118). Let random variable $X$ be a function from the set of all possible outcomes of a specific gamble, denoted by $G_X$, to the set of real numbers. Let each value of function $X$ be the quantity of an object (e.g., money) to be paid or received by the individual who plays this gamble. When $X$ takes only discrete real values $a_1$, $a_2$, $a_3$ ... with probabilities $p_1$, $p_2$, $p_3$ ..., respectively,

this gamble can also be denoted by $\underset{X}{G} = p_1 a_1 \oplus p_2 a_2 \oplus p_3 a_3 \oplus \ldots$ (Varian 1992, pp. 173–76). The current literature claims that under the axioms of EUT, for a *rational* decision-maker, a function $U$ (in a set-theoretic sense) always exists such that $U(\underset{X}{G}) = E[U(X)]$, where $E$ is the expectation operator. Function $U$ is claimed to be such that for any two gambles $\underset{X}{G}$ and $\underset{Y}{G}$ which satisfy the axioms of EUT, one can write $\underset{X}{G} \succ \underset{Y}{G} \Leftrightarrow U(\underset{X}{G}) > U(\underset{Y}{G})$ and $\underset{X}{G} \sim \underset{Y}{G} \Leftrightarrow U(\underset{X}{G}) = U(\underset{Y}{G})$ (note: symbol $\succ$ denotes preference and symbol $\sim$ denotes indifference). As such, $U$ is called the von Neumann–Morgenstern (vN-M) utility function following the latter's 1953 book *Theory of Games and Economic Behavior*, which purports to prove EUT axiomatically in its appendix, whilst relying on its main text for certain definitions and explanations.

The decision-maker can be an individual, a firm or a State. The psychological processes that the decision-maker at an individual or collective level goes through to come to a decision is outside the scope of EUT. The axioms of EUT can be viewed as to what a rational decision-maker either *does* or *should* do; thus, EUT can be seen as a *descriptive* or a *normative* theory of behaviour. EUT was originally devised for gambles with objective probability; Savage (1954) extended it to gambles with subjective probability. Therefore, EUT is currently claimed to be applicable to many real-world risky or uncertain situations, including interactive decision-making scenarios envisioned in game theory, in relation to which EUT is of foundational significance. EUT is claimed to predict or explain rational behaviour, based on the decision-makers' assessments of their prospects, given all the readily available free information to them.

The vN-M book *Theory of Games and Economic Behavior* was first published in 1944 without the axiomatic proof of EUT, as von Neumann and Morgenstern had doubts on the validity of their proof. In the light of Gödel's incompleteness theorems (Gödel 1931), it is not possible to prove the internal consistency of EUT. Nonetheless, it is possible to prove the internal inconsistency of certain theories, and in particular EUT, as EUT is founded on the *vN-M definition of a function* (Von Neumann and Morgenstern 1953, p. 88), which has significant shortcomings by current standards of rigour, and generates the following hereto overlooked internal inconsistencies (note: this is not an exhaustive list):

(a)  If the probability distribution of $X$ has no mean, which can occur as the *axioms of EUT do not rule out any probability distribution of $X$*, no $U(\underset{X}{G})$ can exist.

(b)  If $X$ has a mean, there is no guarantee that $U(\underset{X}{G})$ can exist.

(c)  Leaving out Cases (1) and (2), if $U(\underset{X}{G}) = E[U(X)]$, a *certainty equivalent* $E(X) - \Pi$ for the random variable $X$ (where $\Pi$ is the risk premium of $X$) is claimed to exit such that the decision-maker will be *indifferent* in replacing this random variable with its certainty equivalent or vice versa, i.e., $U(\underset{X}{G}) = E[U(X)] = E\{U[E(X) - \Pi]\} = U(\underset{E(X)-\Pi}{G}) = U[E(X) - \Pi]$. However, no certainty equivalent for any non-degenerate random variable can exist by the set-theoretic definition of the concept of a random variable in statistics.

On the other hand, EUT defines a risk-seeker and a risk-averter in such a way (see Section 5.1 for the definition of these terms) that it becomes impossible for a rational decision-maker to be both a strict risk-seeker and a strict risk-averter *concurrently*, whilst everyday behaviour in the real-world often requires taking risk and seeking protection from it concurrently. For instance, one may choose to have a mortgage for buying a house with a volatile price, aiming to repay the mortgage with one's income, and concurrently seek cover against loss of income by way of having a mortgage payment protection insurance policy. However, under EUT, such behaviour is deemed irrational, revealing the external inconsistency of EUT. Bizarrely, EUT deems the behaviour of this house-buyer rational if he/she seeks no insurance policy. Analogously, under EUT, anyone who chooses to bear the dangers of a new pandemic such as COVID-19, and refuses any protection for his/her health, is regarded as rational. On the other hand, whilst at the same time that

such an individual accepts to bear these dangers, if he/she seeks any protection against this pandemic, he/she is said to be irrational!

The structure of the rest of this article is as follows. Section 2 describes the methodological approach of this paper. Section 3 provides simple cases of the internal inconsistency of EUT to stimulate research interest in this area for the unsuspecting believers in EUT. Section 4 highlights the practical rejection of EUT in risk management in the finance industry. Section 5 generalizes the findings of Section 3 on the internal inconsistency of EUT. Section 6 draws attention to the errors in the existing proofs of the claims of EUT. Section 7 is on the limitations of this paper. Section 8 presents the results of this research and discusses their implications, including an indication of how the problems of EUT can be resolved in a new paradigm. Section 9 discusses the broad implications of this paper. Section 10 concludes. New concepts are defined at the point of their first use.

A critical literature review under five headings is presented in the appendices. Appendix A studies recognized theoretical, empirical and experimental failures of EUT and considers their implications for economic theory. Appendix B addresses the deep problems which utility theory generates for economic theory, by articulating the axiom of scarcity of resources accurately in a formal sense and noting that rational behaviour under utility theory is incompatible with this fundamental axiom; it also points to how these foundational problems in economics/finance can be resolved in a newly emerging paradigm. Appendix C critiques the vN-M definition of a mathematical function on which EUT is founded, and studies ordinal and vN-M utility functions from the Zermelo–Fraenkel set-theoretic perspective. Appendix D brings to light the hidden errors in the current textbook proofs of the existence of a vN-M utility function. Appendix E reviews the notations used for the operation of addition versus the operation of mixture in the proofs of Von Neumann and Morgenstern (1953) and Herstein and Milnor (1953) and demonstrates the contradictions that they generate for these proofs.

## 2. Methodology

Utility theory, including EUT, rests at the foundation of standard economic theory currently. However, it has generated a huge number of puzzles and anomalies (see Appendix A). Whilst ad hoc reasons for these developments have been advanced, the generic cause of these problems has only recently come to light through a rigorous examination of its underlying axiomatic and mathematical foundations (Falahati 2019a). Consistent with the latter approach, the methodology of this paper which focuses on EUT is to scrutinize the underlying axiomatic and mathematical foundations of EUT. This brings to light hereto unaddressed issues on which EUT is founded. In particular, this paper proves that EUT is inconsistent with axiomatic set theory, notes the incompatibility of the axioms of EUT with scarcity of resources and proposes that these are the reasons why EUT is rejected in practice (see Section 4) and in experiments (see Appendix A).

The criteria for the validity of any theory, as a logically constructed representation of part of the real-world, are its internal consistency and the external consistency of its results. For a reviewer of a theory, the examination of its internal consistency normally takes precedence over the examination of its external consistency; however, surprisingly, whilst historically there has been much research on the latter aspect of EUT, highlighting its empirical and experimental failures, virtually no in-depth research has been carried out on its internal consistency. Perhaps, this reflects the widespread belief that EUT is valid as a theory for the perfectly rational specimen; and the observed failures of EUT are attributable to the *bounded* rationality of the ordinary human being (Simon 1955). A view which is reinforced in the light of the high esteem in the existing literature for John von Neumann (1903–1957), appearing as an unerring mathematical genius, on account of his foundational contributions to various mathematical fields (Macrae 1992), one of which was axiomatic set theory. Incongruously, in the latter area, von Neumann's contribution was seriously deficient, in particular in defining the concept of *function*, and left Paul

Bernays (1888–1977) and Kurt Gödel (1906–1978) to correct his oversight, leading to von Neumann–Bernays–Gödel set theory (Hamilton 1982, pp. 115–33, 145–55).

Alas, von Neumann used the same deficient definition of function in his work on EUT (Von Neumann and Morgenstern 1953, p. 88). However, the impact of this error on the proof of the existence of vN-M utility functions has never been explored hereto. This represents a gaping hole in the history of research on EUT. Therefore, the primary focus of this paper is on the examination of the mathematical integrity of the expected utility theorem leading to $U(\underset{X}{G}) = E[U(X)]$ in the light of the set-theoretic definitions of the concepts of *function* and *random variable* in mathematics and statistics. This paper demonstrates that these rigorous definitions are not followed in any proof of this theorem, with fatal consequences for EUT.

### 3. Unnoticed Cases of Internal Inconsistency of EUT

In the following three cases, for all the gambles studied, the random variables generating the outcomes of gambles lead to monetary payments or receipts by individuals who play the gambles, and *whilst the axioms of EUT hold by assumption, yet EUT fails.*

**Case 1.** *Where for gamble $\underset{X}{G}$, the probability distribution of X does not have a mean. This makes it impossible for any vN-M utility function $U(\underset{X}{G})$ to exist, contrary to the prediction of EUT. For instance, let the decision-maker's utility function be an exponential one: $U(X) = (1 - e^{-aX})/a$, as used in standard textbooks, where e is the Euler's number, and $a = 1$, $a = 0$ and $a = -1$ represent risk-averter, risk-neutral and risk-seeker behaviour, respectively, under EUT. The axioms of EUT put no restriction on the characteristics of the random variable X generating the outcomes of gamble $\underset{X}{G}$; hence, X can have a Cauchy probability distribution which has no defined mean, variance or higher moment. Given that no $E(X)$ exists in this case, no $E[U(X)]$ and no vN-M utility function $U(\underset{X}{G}) = E[U(X)]$ can exist either.*

**Case 2.** *Where random variable X has a continuous probability density function f with a finite mean, a and b are finite real numbers, and for gamble $\underset{X}{G}$, $U(\underset{X}{G})$ is an improper integral in the form $U(\underset{X}{G}) = \int_a^\infty U(X)f(X)dX$, $U(\underset{X}{G}) = \int_{-\infty}^b U(X)f(X)dX$ or $U(\underset{X}{G}) = \int_{-\infty}^\infty U(X)f(X)dX$ such that $U(\underset{X}{G})$ does not exist. Geweke (2001) and Yoon (2004) provide examples of such failures of vN-M utility functions; however, they fail to note that these represent internal contradictions for EUT; and they do not explain the cause of these contradictions or offer any remedy for them.*

**Case 3.** *Assuming the axioms of EUT lead to the existence of a multivariate vN-M utility function u, where the probability distribution of each random variable $x_i$ has a finite mean for $i = 1, 2, 3, \ldots$, a certainty equivalent $E(x_i) - \pi_i$ for $x_i$ with risk premium $\pi_i$ is claimed to exist (Kihlstrom and Mirman 1974) such that $E[u(x_1, x_2, x_3, \ldots)] = u[E(x_1) - \pi_1, E(x_2) - \pi_2, E(x_3) - \pi_3, \ldots)]$. Paroush (1975, p. 283) and Duncan (1977, p. 896) recognize that this situation leads to a multiplicity of certainty equivalents for each random element of multivariate wealth derived from the same utility function for the same individual. The existing literature fails to notice that this generates a multiplicity of preference-orderings of the same elements of wealth, each of which can contradict the other, contrary to EUT, which implicitly assumes one and only one preference-ordering of the elements of wealth, hence an internal contradiction for EUT.*

For example, let the decision-maker's vN-M utility function be $u$ of the Cobb–Douglas form, as in Duncan (1977, p. 896): $u(x_1, x_2) = x_1 x_2$, where $x_1$ and $x_2$ are random monetary amounts, each with a finite mean, hence $E[u(x_1, x_2)] = E(x_1 x_2)$. Let the certainty equivalents of $x_1$ and $x_2$ be $E(x_1) - \pi_1$ and $E(x_2) - \pi_2$, respectively, such that $E[u(x_1, x_2)] = u\{[E(x_1) - \pi_1], [E(x_2) - \pi_2]\} = [(E(x_1) - \pi_1)][E(x_2) - \pi_2)]$. Given $[E(x_1) - \pi_1][E(x_2) - \pi_2)] = E(x_1)E(x_2) + \pi_1\pi_2 - \pi_1 E(x_2) - \pi_2 E(x_1)$, and $[E(x_1 x_2) = [(E(x_1) - \pi_1][E(x_2) - \pi_2)] \Rightarrow E(x_1 x_2) - E(x_1)E(x_2) = \pi_1\pi_2 - \pi_1 E(x_2) - \pi_2 E(x_1)]$, by letting $\sigma_{12}$ to be the covariance of $x_1$ and $x_2$, and noting that $\sigma_{12} = E(x_1 x_2) - E(x_1)E(x_2)$, one obtains

$\sigma_{12} = \pi_1 \pi_2 - \pi_1 E(x_2) - \pi_2 E(x_1)$. Duncan (1977, p. 896) misstates the latter equality in the form $\sigma_{12} = \pi_1 \pi_2 - \pi_1 x_2 - \pi_2 x_1$!

Assuming $E(x_1)$, $E(x_2)$ and $\sigma_{12}$ exist and are finite and known, the latter equality is one equation for two unknown risk premia, i.e., $\pi_1$ and $\pi_2$; thus, there can be no unique certainty equivalent for either $x_1$ or $x_2$.

This generates an internal contradiction for EUT. To see this, let $\pi_1 = p$ and $\pi_2 = q$ such that $\sigma_{12} = pq - pE(x_2) - qE(x_1)$ and $p - q < E(x_1) - E(x_2)$, hence $E(x_2) - q < E(x_1) - p$. It follows that the certainty equivalent of $x_1$ is greater than the certainty equivalent $x_2$, and thus for gambles $\underset{x_1}{G}$ and $\underset{x_2}{G}$, one can write $\underset{x_1}{G} \succ \underset{x_2}{G}$. On the other hand, one can let $\pi_1 = m$ and $\pi_2 = n$ such that $\sigma_{12} = mn - mE(x_2) - nE(x_1)$ and $m - n > E(x_1) - E(x_2)$, hence $E(x_2) - n > E(x_1) - m$.

It follows that the certainty equivalent of $x_1$ is less than the certainty equivalent $x_2$, and thus for gambles $\underset{x_1}{G}$ and $\underset{x_2}{G}$, one can write $\underset{x_2}{G} \succ \underset{x_1}{G}$, hence a contradiction for EUT. Let us note that any single random variable $x$ can be thought of as a product of two random variables such that $x = x_1 x_2$; hence, this internal contradiction can arise for univariate *and* multivariate random wealth.

## 4. EUT versus Risk Management in Practice in the Finance Industry

Friedman and Savage (1948) observe that contrary to EUT, the same decision-maker can be a risk-seeker and a risk-averter at two different levels of wealth (see Appendix A). However, they do not realize that a decision-maker can also be a strict risk-seeker and a strict risk-averter *concurrently*, i.e., at the same level of wealth. This is easily seen in the finance industry, as the following explains.

For example, when a bank engages in covered interest rate arbitrage to exploit the interest differential on deposits in two different currencies, it acts as a strict risk-seeker (by exposing the bank to currency fluctuations) *and* a strict risk-averter (by using forward contracts to protect the bank from currency fluctuations); however, such behaviour is irrational under EUT.

The empirical failure of EUT can be seen quite clearly in relation to the management of risk by insurers, an industry which Bernoulli (1954, p. 30) refers to, in justifying his belief in an early version of EUT. For, under EUT, a rational decision-making firm cannot be a specialist insurer (e.g., a marine insurer) as a strict risk-seeker *and* also obtain reinsurance as a strict risk-averter against claims it may not be able to afford, contrary to everyday practice in the insurance/reinsurance industry. The rationale for this industry practice is that specialization gives the insurer a competitive advantage against non-specialist insurers, and reinsurance can curtail the cost of policyholders' claims to a limit which can help the specialist insurer avoid insolvency. Moreover, under EUT, it is not rational for this specialist insurer (as a strict risk-seeker) to invest (as a strict risk-averter) the premia it receives from its policyholders in risk-free bonds rather than risky assets (e.g., equities) to cover its cost of expected claims, contrary to the normal investment practice in this industry. The rationale for this industry practice is that investing in risk-free bonds rather than equities reduces the insolvency risk of the specialist insurer, e.g., when a marine insurer concurrently faces a tsunami of claims and the collapse of equity markets as a result of a sudden war in the seas.

## 5. Irreparably of EUT

This section demonstrates the self-contradictory implications of rational behaviour under EUT in the disciplines of economics/finance and mathematics/statistics.

### 5.1. Implications of EUT for the Standard Paradigm in Economics/Finance

Let us note that Cases 1 and 2 of Section 3 cannot be amended by making the vN-M utility function bounded as no vN-M utility function with a defined mean can exist in these cases. On the other hand, the axioms of EUT do not make the vN-M utility function bounded. To avoid the problems encountered in these cases, one needs to add to the axioms of EUT a *new* axiom: *the size of the actual or expected outcomes of the gambles, whether simple or*

*compound, must never approach positive or negative infinity, when measured in non-infinitesimal units.* The justification in economics and finance for this restriction is the axiom of scarcity of resources (see Appendix B), which requires that at any date, the total quantity of each scarce resource is finite, and the total number of all the different types of distinct scarce resources is also finite. This axiom requires that as a *valid* claim against scarce resources, the quantity of money, whether in commodity form (e.g., gold) or in fiat form (e.g., $), must be finite at any date, as otherwise it will cause unbounded inflation. Therefore, one cannot pay or receive an actual or expected infinite monetary outcome of any gamble, and thus such a gamble is never played. Incidentally, this new axiom for EUT removes the St. Petersburg paradox, the resolution of which was the historical motivation for the development of an early version of a vN-M utility function (Bernoulli 1954).

For supporters of EUT, this new axiom is unwelcome, as it eliminates applications of EUT to gambles with unbounded probability distributions of their outcomes, e.g., the normal distribution, which are relied on in financial economics. Supporters of EUT argue that if the utility function is bounded, this objection does not hold; however, they ignore the fact that in no economy which admits scarcity of resources, an infinite quantity of money (in commodity or fiat form) can ever exist; hence, the St. Petersburg paradox will not be offered in such an economy.

Let us presume that this new axiom is complied with, and the axioms of EUT (Jehle and Reny 2011, pp. 111–13) hold for gamble $\underset{Z}{G}$ and lead to the existence of $U$ as a *strictly increasing* vN-M utility function for a decision-maker with sure monetary wealth $M$, where $Z$ is the random monetary outcome of $\underset{Z}{G}$ with a known probability distribution. Then, this decision-maker is claimed to be a strict *risk-averter* if for him/her $U[E(M+Z)] > E[U(M+Z)]$, and he/she is claimed to be a strict *risk-seeker* if for him/her $U[E(M+Z)] < E[U(M+Z)]$, and he/she is claimed to be *risk neutral* if for him/her $U[E(M+Z)] = E[U(M+Z)]$. Thus, for strict risk-averters, $U$ is strictly concave, and for strict risk-seekers, $U$ is strictly convex, whilst for risk neutral decision-makers, $U$ is strictly linear. This is where EUT does not define the concept of risk *per se* such that the meaning of attraction towards it i.e., risk-seeking, or repulsion from it i.e., risk-aversion or indifference towards it i.e., risk-neutrality, can be unambiguously derived from the definition of risk. This is where the decision-maker is implicitly assumed to be indifferent towards gambling *per se*, as gambling per se generates no utility or disutility for decision-makers under EUT (Von Neumann and Morgenstern 1953, pp. 629–30)!

Supporters of EUT seek to justify the trading of a lottery (e.g., gamble $\underset{Z}{G}$ with random monetary prize $Z$) at a fixed price by deriving an equivalence relationship between the random prize of the lottery and the certainty equivalent of this random prize as follows: Consider a decision-maker with a strictly increasing vN-M function $U$ with respect to each of its variables. Following the Jensen inequality and Pratt (1964), if this decision-maker has the sure monetary wealth $M$ and the non-degenerate random monetary outcome $Z$ with mean $E(Z)$, arising from his/her right to play gamble $\underset{Z}{G}$, one can define *risk premium* ${}_M h_Z$ such that

$$U(\underset{M+Z}{G}) = E[U(M+Z)] = E\{U[M+E(Z) - {}_M h_Z]\} = U(M+E(Z) - {}_M h_Z), \quad (1)$$

or

$$U(\underset{M+Z}{G}) = E[U(M+Z)] = E\{U[M+E(Z) - {}_M h_Z]\} = U(\underset{M+E(Z) - {}_M h_Z}{G}). \quad (2)$$

The sure amount $E(Z) - {}_M h_Z$ is claimed to be the *certainty equivalent* of the random amount $Z$ when the remainder of the decision-maker's wealth is the sure amount $M$, in the sense that he/she will be *indifferent* in replacing the random amount $Z$ with the certain amount $E(Z) - {}_M h_Z$. $E(Z) - {}_M h_Z$ is thus claimed to be the minimum selling (or maximum buying) price that the decision-maker requires to give up (or acquire) the right to play $\underset{Z}{G}$. If the decision-maker was a strict risk-averter, $U$ would be concave and ${}_M h_Z > 0$, and if the

decision-maker was a strict risk-seeker, $U$ would be strictly convex and $_M h_Z < 0$, and if the decision-maker was risk-neutral, then $U$ would be linear and $_M h_Z = 0$. Nonetheless, the foregoing new axiom does not remove the objections to EUT that arise under Case 3 in Section 3, which has a much wider scope than Cases 1 and 2. Indeed, as the following proposition proves, rational behaviour under EUT is incompatible with the axiom of scarcity of resources (defined in Appendix B), and in particular, it is incompatible with the standard paradigm in economics and finance.

**Assumptions of the proposition:** In any competitive, efficient and frictionless economy (CEFE), by definition, free lunches are fully exploited. In the standard CEFE, by assumption, a spot transaction takes zero length of time, i.e., it occurs *timelessly* at a date of zero length on a timeline. In contrast, in the CEFE of the new paradigm (see Appendix B), by assumption, a spot transaction takes a positive length of time to occur, however small that length may be.

　　Consider a standard CEFE in which the axioms of EUT hold for all decision-makers in respect of all gambles, including gamble $\underset{Z}{G}$, where $Z$ is a positive non-infinitesimal random amount of money (in the fiat form, e.g., $, or in the commodity form, e.g., gold). In addition, by assumption, there are always strict risk-averter, strict risk-seeker and risk-neutral gamblers in this CEFE, and strict risk-averters have the property right to play gambles, whilst strict risk-seekers wish to acquire this right. Let us assume that at date $d$ for strict risk-averter A, the certainty equivalent of $Z$ is $E(Z) - h$, and for strict risk-seeker B, the certainty equivalent of $Z$ is $E(Z) + k$, and for risk-neutral gambler C, the certainty equivalent of $Z$ is $E(Z)$, where $h > 0$ and $k > 0$ are sure amounts of money. Further, by assumption, strict risk-averter A has the property right to play gamble $\underset{Z}{G}$, and strict risk-seeker B wishes to acquire this right.

**Proposition.** *In a CEFE of the standard paradigm where spot transactions take place timelessly and by assumption all risk attitudes always exist, rational behaviour as defined by EUT generates arbitrage opportunities, leading to a free lunch and a money pump.*

**Proof.** Let us note that at date $d$, risk-neutral gambler C has an arbitrage opportunity in terms of the standard paradigm in economics and finance, as he/she can pay A the amount $E(Z) - h$ to acquire the right to play gamble $\underset{Z}{G}$ from A and receive $E(Z) + k$ from B for transferring his/her right to play gamble $\underset{Z}{G}$ to B. Consequently, these transactions occur without any loss of utility for A and B and with C ending up with a free lunch of $k + h > 0$ at date $d$. Clearly, within the standard paradigm of economics and finance in this CEFE, where spot transactions take place timelessly, such arbitrage transactions can be repeated at an infinite number of dates within a finite period to generate a money pump or an infinite quantity of gold for C. These trades violate the axiom of scarcity of resources, whilst they reflect rational decision-making under EUT! This is despite the fact that these certainty equivalents, which were derived earlier in this subsection, were based on the presumption that the new axiom, which was introduced to ensure consistency with the axiom of scarcity of resources, could hold under EUT. It is now clear that this new axiom cannot hold under EUT, which means that EUT is incompatible with the axiom of scarcity of resources! □

*5.2. Implications of EUT for Mathematics/Statistics*

　　It is the *continuity* axiom of EUT (e.g., Ingersoll 1987, p. 10; Varian 1992, p. 174; Jehle and Reny 2011, p. 100) which leads to the concept of certainty equivalent of a random variable. For, according to this axiom, if $a, b$ & $c$ are objects of choice for which $a \succ b \succ c$, then there will be a unique $\mu$ such that $0 \leq \mu \leq 1$ and $b \sim \mu a \oplus (1 - \mu)c$. This is where, by letting random variable $X$ of gamble $\underset{X}{G}$ take values $a$ and $c$ with probabilities $\mu$ and $1 - \mu$, respectively, one obtains $\underset{X}{G} = \mu a \oplus (1 - \mu)c$ by definition; and by letting $a, b$ & $c$ be sure

monetary amounts, it follows from $b \sim \mu a \oplus (1 - \mu)c \Rightarrow \underset{b}{G} \sim \underset{X}{G}$. Thus $b$ becomes the certainty equivalent of random variable $X$.

However, this is contrary to the definition of a *random variable* (Doob 1996, p. 590; Fristedt and Gray 1997, p. 11) as a measurable function which assigns a single numerical value to each possible outcome of an experiment with a well-defined set of outcomes with no predictable order of realization of each one of the individual outcomes when they may not be the same outcomes. Thus, a non-degenerate random variable cannot have a *unique* specific value, but it will have a set of values. On the contrary, the existence of a certainty equivalent for a non-degenerate random variable leads to a unique specific value for the random variable and implicitly requires the existence of an outcome which is equivalent to each one of the *different* outcomes of such an experiment. This can be deduced formally from the definition of the certainty equivalent as in the following:

Consider gamble $\underset{Z}{G}$ in Section 5.1, where by the definition of the concept of certainty equivalent, the decision-maker must be *indifferent* in replacing $Z$ with $E(Z) - {}_M h_Z$ in his/her utility function $U$, and he/she must be also *indifferent* in replacing $E(Z) - {}_M h_Z$ with $Z$ in $U$. It follows from Equation (1) that whatever the decision-maker's risk attitude may be $E[U(M + Z)] = U[M + E(Z) - {}_M h_Z]$. Replacing $Z$ with $E(Z) - {}_M h_Z$ on the left side of the latter equality and replacing $E(Z) - {}_M h_Z$ with $Z$ on the right side of it yields $E[U(M + E(Z) - {}_M h_Z)] = U[M + Z]$ or $U[M + E(Z) - {}_M h_Z] = U[M + Z]$. Given that $U$ is a one-to-one function, then $E(Z) - {}_M h_Z = Z$, which is absurd! If every random variable had a certainty equivalent, there would be no need for the discipline of statistics.

Supporters of EUT presume that since a lottery ticket can be traded, like any other commodity, at a fixed price, therefore its random prize must have a certainty equivalent, such that both traders in such an exchange will be *indifferent* between the fixed price and the random prize of the lottery. Let us note that this view considers the only source of value of a gamble to traders is its random prize, and it overlooks any positive or negative value that *gambling per se* may have. Moreover, let us note that trade takes place between two parties, who exchange one object with another, when both parties think that they will be *better-off*, rather than being *indifferent*, in doing so. For, if the parties are indifferent, then the exchange will be reversible and can be annulled. Hence, it is *not* possible to conclude from this assumed *indifference* that such trade *must* occur between these two parties. Therefore, there is no compelling logic for equating the price of a lottery ticket to such a derived certainty equivalent of its random prize.

## 6. Why Existing Proofs of EUT Are Fallacious

The previous sections raise the question as to what is wrong with the existing proofs of existence of vN-M utility functions. I have touched on some of these reasons already. I address them thoroughly in Appendices C–E, an overview of which follows here.

Appendix C provides, by current standard of rigour in mathematics which employs the concept of function in its set-theoretic sense, the *criteria to judge* the mathematical integrity of any claim to the proof of the existence of any utility function. In doing so, it disproves ordinal and expected utility theories without challenging their axioms, and only by relying on the set-theoretic definition of a function.

The proofs of existence of vN-M utility functions in current textbooks (e.g., Varian 1992, pp. 172–76; Jehle and Reny 2011, pp. 97–118) overlook the errors which Appendix C brings to light, and are proved to be self-contradictory by a theorem in Appendix D.

Von Neumann and Morgenstern (1953) and Herstein and Milnor (1953) use the same notation for the operations of *addition* and *mixture*, Appendix E brings to light the self-contradictory implications of this conflation and thus rejects both these proofs. Herstein and Milnor (1953) try *not* to contradict explicitly *either* the vN-M *or* the post-set-theoretic concept of a function; thus they do not define what they mean by a mathematical function, which is fundamental to their proof; hence their analysis is deficient in this crucial respect. Moreover, the axioms which Herstein and Milnor (1953, pp. 292–95) impose on their mixture sets generate their Theorem 6. The latter supports, and indeed upholds, the

vN-M axiom of continuity, which, as discussed in Section 5.2, leads to EUT becoming self-contradictory.

Remarkably, neither Herstein and Milnor (1953) nor Von Neumann and Morgenstern (1953) show any awareness of the failures of EUT in such simple cases as those in Section 3, and thus they make no attempt to remedy these failures. In contrast, to justify the need for the axiom of continuity in ordinal utility theory, to which all these authors subscribe, they are perfectly content with just one counterexample, i.e., the well-known case of lexicographic preferences. This paper points out that *EUT cannot be valid with or without the axiom of continuity*, as EUT violates the axiom of scarcity of resources and it lacks mathematical integrity, highlighted in Section 5 and Appendices B–E.

## 7. Limitations

The description of the newly emerging paradigm as a new overarching economic theory which resolves long-standing puzzles and accords with the real-world, discussed in Appendix B, is quite condensed. To engage in an expanded description would make this article too long. Nonetheless, there is a need to elucidate the new paradigm further and show in a separate article in the future the extent it can go to resolve the extant anomalies of the currently dominant paradigm.

## 8. Results and Discussion

This article notes that contrary to the definition of a random variable, a vN-M utility function is claimed to transform a non-degenerate random variable into its certainty equivalent, and brings to light the internal contradictions that this implies for EUT. On the other hand, it explains that there is no compelling logic for equating the price of a lottery ticket to such a certainty equivalent. This paper articulates precisely for the first time the axiom of scarcity of resources in its primary form, and it points out that the axioms of utility theories, including EUT, are not consistent with scarcity of resources (e.g., Section 5.1); hence they cannot be rational. It highlights why certain persistent observed individual behaviour in the face of risk and certain persistent observed behaviour in risk management in the finance industry, including the insurance industry, are economically justifiable, contrary to EUT which regards such behaviour irrational. The findings of this paper are relevant to the development of any alternative theory to EUT and explain why attempts to construct new functions which can describe risky/uncertain economic behaviour in non-EUT research studies have not achieved sweeping resolutions of existing anomalies. In contrast, a condensed description of a new paradigm is provided where *very many major* behavioural puzzles are resolved without such a function. These include long-standing puzzles such as St. Petersburg paradox, instant endowment effect, asymmetric valuation of gains and losses, intransitivity of preferences, the reason for the existence of the firm and the explanation on how the profit of the firm emerges and it is financed in the CEFE of the new paradigm.

## 9. Broad Implications of This Paper

The history of mathematics, like all other intellectual disciplines, illustrates that it is possible for a new generation to identify errors in the works of an earlier generation. The reason why such polymaths as Daniel Bernoulli and John von Neumann could not see the internal inconsistencies of EUT disclosed here is that in common with many mathematicians of their eras, for a very long time, mathematicians did not employ rigorous definitions of the mathematical concepts of function and random variable. The evidence for the latter facts is seen in Kleiner (1989, p. 284) and in Von Neumann and Morgenstern (1953, p. 88). It is in this context that Doob (1996, p. 586) notes, 'The mathematization of probability required new ideas, and in particular required a *new* approach to the idea of . . . a *function*' (emphasis added).

Doob (1996, p. 588) also notes the observation of Max Planck (1858–1947) on the power of received wisdom, where Planck states, 'A new scientific truth does not triumph by

convincing its opponents and making them see the light, but rather because its opponents eventually die, and a new generation grows up with it.' This can be emphatically so when the new idea challenges the validity of received wisdom. I trust that scientists have learnt from history and that this gloomy prediction of Planck will not be applicable to new scientific truths found in modern times. However, it is hard to be optimistic, as the sad and the striking finding of this research is that too many intelligent people and their followers, over many generations, can be mesmerized by what appears to them as an elegant theory, such as EUT, based on a non-rigorous justification. The aesthetic attraction of such a theory and halos of its promotors in the eyes of its followers seem to motivate repeated propagation of it, regardless of its internal and external inconsistencies. Hence, such a myth, dressed up as science, is spread widely, and its repetition to the naive makes it believable to them, as long as their views are left unchallenged.

The problem becomes quite serious when this theory is routinely taught in educational institutions as part of a standard curriculum, and thus mythology masqueraded as science dominates scientific discourse. This is where any rejection of the extant theory by the avant-garde, however well substantiated, will be opposed by the laggard, given their vested interests in the status quo, as the latter fear losing respectability for their research and teaching materials. This gives rise to a duel between the mythology supported by the dominant paradigm and the challenges of the avant-garde to it. The shift to a new paradigm for science can come if a new generation is able to sift through *independently* and *freely* all relevant received wisdom carefully, impartially and thoroughly and verify the new scientific truth.

## 10. Conclusions

This paper demonstrates that EUT is founded on pre-set-theoretic understandings of what a mathematical function and a random variable is and that it is incompatible with modern mathematics/statistics and it is also incompatible with the foundational concept of scarcity in economics/finance. For, EUT generates internal contradictions in mathematics/statistics and economics/finance. A new paradigm which overcomes these problems and resolves very many anomalies of the standard paradigm in microeconomics, financial economics and macroeconomics is presented. To help this new paradigm be more widely recognized, there is a need to elucidate it more and to expand it by demonstrating its potential for resolution of any more remaining anomalies in the future.

**Funding:** This research received no external funding.

**Institutional Review Board Statement:** Not applicable.

**Informed Consent Statement:** Not applicable.

**Acknowledgments:** The author is grateful for the comments of three anonymous reviewers, and for communications of the Academic Editor Pawel Ziemba, and for Bessie Chen's efforts in the production of this article. The author dedicates this article to the memory of his wonderful brothers: Reza, Kamal & Sadegh.

**Conflicts of Interest:** The author declares no conflict of interest.

## Appendix A. Already Recognized Failures of EUT and Their Implications

Falahati (2019a) notes three errors in the proof of ordinal utility theory; this leads to the rebuttal of EUT which relies on ordinal utility theory. Yoon (2004) notes that EUT fails where constant relative risk aversion is assumed and the decision-maker's endowment follows a stochastic unit root process. Geweke (2001) provides further examples in this connection. However, neither Yoon nor Geweke point out explicitly that these failed cases represent internal inconsistencies for EUT, whilst they implicitly note that as none of the axioms of EUT are violated, hence EUT must not fail.

Given the foregoing internal inconsistencies of EUT, identified since at least 2004, it is no surprise to note its external inconsistencies, which have been known since at least

1948 (see below). Surely, if these internal inconsistencies were discovered earlier, a huge amount of talent would not be needed to demonstrate the external inconsistencies of EUT. Friedman and Savage (1948) note that the same individual can be a risk-seeker and a risk-averter at two different levels of wealth, contrary to EUT which assumes his/her preferences are not dependent on his/her levels of wealth. In contrast, Section 4 points out that a decision-maker can be a risk-seeker and a risk-averter concurrently, i.e., at the same level of wealth, contrary to EUT. Moreover, repeatable experiments show that EUT does not work well either normatively (e.g., Allais 1953) or descriptively (e.g., Tversky and Kahneman 1986). Indeed, researchers have found almost no empirical evidence for the existence of any vN-M utility function (Chiappori et al. 2019).

Therefore, behavioural scientists are quite understandably moving away from EUT; thus a special issue of Development Review in March 2008, dedicated to 'Current Theories of Risk and Rational Decision Making', made no reference to EUT (Reyna and Rivers 2008). Many theories which rely on EUT, e.g., game theory, are found to have their results generate persistent anomalies and external inconsistencies, which reduces their credibility. Inevitably, serious scholars must confront these inconsistencies and endeavour to remove them, hence the need for a new paradigm such as the one described in Appendix B, which has already resolved very many problems of the standard paradigm in microeconomics, financial economics and macroeconomics (Falahati 2019b, 2019c).

## Appendix B. Scarcity of Resources versus Utility Theory

This appendix articulates the axiom of scarcity of resources clearly and explains the incompatibility of this axiom with utility theory, and concludes with a brief description of a new paradigm which can resolve this problem. Robbins (1932, p. 15) states succinctly:

'Economics is the science which studies human behaviour as a relationship between ends and *scarce means which have alternative uses*' (emphasis added).

For this definition to be complete, the concept of scarcity of means/resources, as the fundamental axiom of economics, must be defined *formally* (i.e., in its precise, full, irreducible form and exclusive sense such that it has only *one* meaning) to put economic theory on a firm scientific foundation. However, despite its importance, the existing literature fails to provide a satisfactory formal definition of scarcity, as Falgueras-Sorauren (2017) notes. This paper seeks to do so, starting with preliminary definitions:

A *date* is taken to be a point of no length on a timeline. The phrase 'A and/or B' in this appendix means one of the following three things: both A and B, only A or only B. I refer to the *owning* and/or *owing* of an object as *engagement with* that object. Once an individual engages with an object, as a consequence, he/she can subsequently do different things with it, e.g., consume it, invest it, trade with it, give it away or throw it away, each of which can have a different effect on his/her wealth and feeling of *welfare* (i.e., well-being). Thus, I draw a distinction between initial *engagement* with an object and the subsequent *consequence of engagement* with it. This is where it is recognized that for an individual, *to engage* with an object *per se* can generate a *welfare-gain* (i.e., a feeling of being better off) or a *welfare-loss* (i.e., a feeling of being worse off), as a result of acquiring or losing property rights over it. Moreover, it is recognized that these feelings of the individual are separate from his/her subsequent feelings of welfare for the different activities which he/she may do with those objects.

Leaving out fiat money, which is defined later in this appendix, a *good* for an individual is an object which he/she strictly prefers engaging than not engaging with, in which case, engagement with it generates a net welfare-gain for him/her. Further, a *bad* for an individual is an object which he/she strictly prefers not engaging than engaging with, in which case, engagement with it generates a net welfare-loss for him/her. In addition, the loss of a good (not caused by owner's own consumption) generates a welfare-loss, and the reduction of a bad generates a welfare-gain, for the individual.

By assumption, *physical* objects are quantifiable and countable at any date, and so are living species as physical objects. The set of physical objects at any date is finite. At

any *date*, each *distinct scarce resource* is a *finite* quantity of a physical good (produced by Man or Nature) which is available to living human beings, and on which it is possible for human beings to have private or public property rights. *The axiom of scarcity of resources, as a universal axiom underpinning economics as a scientific discipline, requires that the proper set of all scarce resources which exist for all living human beings at any date be finite.* This set is obviously not necessarily going to be the same at all times i.e., fixed for all dates. Moreover, as most species live interdependently, humankind has to share the use of at least some of these scarce resources with other living species.

A *service* requires physical goods as well as labour-time and/or machine-time to be carried out, and it is regarded as an object in this context and thus can be a good. *Non-physical* goods such as knowledge require physical goods and time for their production, distribution and application. Each of the latter activities represents a service. The axiom of scarcity of resources constrains the provision of services at each date.

**Money:** Under the axiom of scarcity of resources, the quantity and number of any physical good (e.g., gold) used as a means of exchange is finite at any date. Moreover, *fiat money*, being an agreed means of exchange in the economy between the State and the citizenry, must provide a valid claim against scarce resources by definition; hence the State must ensure that its total quantity is finite at any date to avoid unbounded inflation. For the same reason, there can never be an infinite number of valid fiat currencies.

**Axioms of utility theories versus axiom of scarcity of resources:** The objects of choice take the form of goods in ordinal utility theory and gambles in EUT. The first axiom of ordinal utility theory (and EUT) is completeness of the preferences over objects of choice, which embodies the *axiom of reflexivity*. Under the axiom of reflexivity, a decision-maker can replace an object with its perfect substitute at any date, i.e., *timelessly* without any change in his/her utility. Moreover, he/she can do so infinitely many times timelessly, without causing any change in his/her utility. *Falahati (2019a, p. 33) proves that the axiom of reflexivity is inconsistent with the axiom of scarcity of resources.*

The proof of Falahati (2019a, p. 33) that ordinal utility theory, on which EUT relies, contradicts the axiom of scarcity of resources implies the need for a new behavioural foundation in economics, leading to a new paradigm. The other findings of Falahati in the latter article make it impossible to have any function representing an individual's preference-orderings over objects of his/her choice, be it a set of goods or gambles. This suggests that tinkering with utility functions, as seen in non-expected utility theories (Machina 2008a), is unlikely to resolve the almost endless problems of EUT.

**Characteristics of the new paradigm:** Falahati (2019b, pp. 120–23) overcomes the foregoing deep problems of ordinal and expected utility theories in a new paradigm of a competitive, efficient and frictionless economy (CEFE), where, inter alia, there is no utility function and each individual can have more than one preference-ordering over their objects of choice. The following explains key characteristics of the new paradigm:

*Preferences over objects of choice*: In this new paradigm, *choice-ordering* of objects is distinguished from *trade-ordering* of objects. Choice-ordering of objects (e.g., for consumption purposes), which tends to be stable, is made when preferences over objects are decided without the need for trading any of them. In contrast, trade-ordering of objects occurs when the exchange of these objects becomes necessary. In the latter case, traders, as decision-makers, implicitly reveal their trade-ordering by *monetary bids and offer quotes* for each object. This is where Falahati (2019b, p. 122) assumes that at each date:

> 'each individual, having taken account of all available information, can assign a monetary value to each of his/her sources of potential welfare gain or loss arising from his/her acquisition or deprival of each object (e.g., a good or a gamble).' [in a continuously open market]

This assumption determines the individual's *trade-orderings* of his/her objects of choice. In doing so, traders take account of their own solvency status; hence *trade-orderings are wealth-dependent and are revised in the light of any new information*. Each trader can determine his/her trade-preferences as a buyer and a seller with *different* bid and offer

monetary quotations for each object at the same date, which will reflect each trader's trade-orderings separately as a buyer and as a seller. Traders can readjust their trade-orderings in this continuously open market by revising their offer and bid quotations in the light of other traders' quotations, until demanders and suppliers agree with each other's declared quotations for an object; if and when they agree, trade occurs between them in that object. In which case, the quotation for that object will become its *objective price.* If they do not agree, the object remains *illiquid*. Let us note that during this price formation process, while trade-orderings can change, choice-orderings need *not* change, and hence the latter can remain *stable* in the new paradigm.

In the new paradigm, even spot transactions take a non-zero quantity of time to occur (Falahati 2019b, p. 121) in a CEFE, and money in any form is always scarce, and arbitrage opportunities are unbounded, e.g., there can be an infinite number of exchange-traded funds (ETFs) from a finite number of assets, with each ETF trading at multiple prices. Thus, arbitrage can reduce, but *not* eliminate, bid–offer spreads due to the opportunity cost of the scarce capital tied up during arbitrage transactions. This leads to separate bid and/or offer prices for each good or gamble for each trader in the new paradigm. Hence, in the new paradigm, the law of one price in its standard sense, where a market participant can buy *and* sell the same good at the same price at the same date cannot hold. The latter law is called the strong law of one price (Falahati 2019b, pp. 127–29). However, it will still be possible for the same market participant to buy *or* sell the same good or gamble at the same price at the same date, as in the standard paradigm. One can thus assume that the same market participant can buy *or* (not *and*) sell any quantity of the same good at the same date at the same price in the new paradigm. The latter is called the weak law of one price (Falahati 2019b, pp. 127–29), and by assumption, it is upheld in the CEFE of the new paradigm.

*Risk*: The meaning of risk is not clearly specified in the standard paradigm, as explained in Section 5.1. In the new paradigm, any risk borne from playing a gamble reflects a possible net welfare-loss arising from its outcomes (Falahati 2019b, pp. 123–24). Therefore, if no welfare-loss is anticipated from playing a gamble or indeed any other activity, no risk is borne. For example, a free ticket to play a lottery with only positive prizes generates no risk to the player; even if ex ante, the player does not know the prize. This is true as long as this lack of foreknowledge implies no welfare-loss for him/her. As one's net welfare-loss can be another's net welfare-gain, one cannot have homogenous expectations of risk in the new paradigm.

Risk, as defined here, can generate what Slovic et al. (2005) call feelings of hazard for an individual/group. Having defined risk per se in the new paradigm, concepts of attraction to, and repulsion from, risk can be meaningfully derived from it, and thus the risk-seeker and risk-averter concepts will have clear meanings, and will be different compared to those vaguely implied by EUT. For example, in the new paradigm, it is possible for an individual to engage in motor-racing and seek protection from its dangers concurrently, as long as this individual considers that these combined activities will not generate a net welfare-loss for him/her. This behaviour is not admissible in the standard paradigm, as explained in Section 1, whilst the new paradigm can coherently accommodate such behaviour.

The new paradigm recognizes that risk is ubiquitous, and seeking protection from it may not be fully possible on account of scarcity of resources. Moreover, it identifies an inverse relationship between liquidity premia and risk premia (Falahati 2019c) which generates risk-premium rating cycles and macroeconomic swings (Falahati 2019c, pp. 166–76). The latter cycle is known as the underwriting cycle in the non-life insurance/reinsurance industry, which is globally well-known for its historical cyclicality.

In the new paradigm, the probabilities of the outcomes of events on which there is incomplete information and generate such welfare effects which give rise to risk are assumed to be their reasonable expectations, based on the axiomatic interpretation of the concept of probability as a reasonable expectation, initially developed by Cox (1946) and

later improved by Dupré and Tipler (2009). This interpretation of probability helps admit non-repeatable events, whist not contradicting standard probability calculus. Cases, where unique probabilities are not decidable, are outside the scope of this paper.

*The holistic decision-maker*: In the new paradigm, a holistic decision-maker determines for himself/herself the net welfare effect of his/her initial engagement with an object and the immediate consequence of his/her engagement with it. This is where, by assumption, all decision-makers are holistic and every holistic decision-maker is risk-caring (2019b, pp. 123–27) 'in the sense that he/she recognizes the change in his/her existing level of risk, and never chooses to offer and/or bid for an unbounded level of risk. Further, ceteris paribus, he/she requires compensation for accepting a greater level of risk than his/her existing level of risk and is willing to pay an affordable compensation to reduce his/her existing level of risk' (Falahati 2019b, pp. 123–24).

Given that in the new paradigm, a monetary value is assigned to each one of an individual's sources of potential welfare-gain or welfare-loss, including compensation for risk, a subjective monetary measure of the opportunity cost in respect of the net effect on his/her welfare of each of his/her decisions becomes available to the decision-maker. In the new paradigm, every holistic decision-maker seeks to minimize his/her opportunity cost based on his/her perception of the alternative ways of achieving his/her aim of improving his/her welfare, whatever form that aim may take, using the scarce resources in the economy. In all decision-making situations, including interactive scenarios envisioned in game theory, the principle of minimizing opportunity cost in all decision-makings is applicable in the new paradigm, and it replaces all the optimization principles of the standard paradigm, including expected utility maximization hypothesis.

*Institutions:* The CEFE of the new paradigm, unlike the standard paradigm, recognizes explicitly the roles that the State, the Central Bank and the banking system play in of a typical actual economy in modern times (Falahati 2019b, p. 121). This economy, in the sense explained in Falahati (2019b, p. 121; 2019c, p. 159) and for the reasons given there, is frictionless whilst it accommodates many realistic features of the real-world which help analyse the essential characteristics of an actual economy, features that are not admitted in the standard paradigm. This leads to more realistic theories, including the loan input-output cycle theory of banking (Falahati 2019c, pp. 159–66) which improves current understandings of how the banking system works, as it explains how an economy with a banking system, compared to an economy without a banking system, creates *extra loans* and *new* money, whilst it can also generate widespread customer funding gaps (Falahati 2019c, pp. 182–83) endogenously. Thus, it can generate booms and busts, leading to systemic banking crises, as it did globally in 2007/8.

**Resolution of puzzles of the standard paradigm in the new paradigm:** I present the resolution of certain major puzzles of the standard paradigm in the new paradigm under two broad headings; initially focussing on puzzles in microeconomics and later on puzzles in macroeconomics:

*Breakdown of explicit and implicit axioms of EUT*: The axiom of reflexivity is implicit in the axiom of substitution of EUT, where the decision-maker is assumed to be indifferent if each outcome of a gamble is replaced with its perfect substitute. Given that the independence axiom of EUT can be derived from combining the substitution and reduction of compound lotteries axioms (Jehle and Reny 2011, p. 101), it is clear that the independence axiom *cannot* hold in a world of scarcity of resources either. This explains the findings of many authors (e.g., Allais 1953) who show that the independence axiom is the cause of the experimental failures of EUT. Hence, functions which purport to represent preference-orderings, and incorporate the axiom of reflexivity in EUT or non-EUT, contradict the axiom of scarcity of resources implicitly.

In the absence of the cost-free theory of arbitrage in the CEFE of the new paradigm (Falahati 2019b), there is no compelling logic to uphold the axiom of transitivity of preferences (Tversky 1969). This makes this new paradigm more realistic compared to the CEFE of the standard paradigm. Falahati (2019b, pp. 124–26) explains that this resolves

behavioural puzzles such as instant endowment effect, asymmetric valuation of gains and losses, and preference reversals.

Similarly, the Dutch Book argument need not hold in the CEFE of the new paradigm. This removes the objection of incoherency (Berger 2006, p. 395) to objective Bayesian procedures for statistical analysis with non-constant prior probability distributions, and makes it possible to admit the reasonable expectation interpretation of probability in the CEFE of the new paradigm.

It is therefore clear from the foregoing critical review of EUT that virtually all the axioms of EUT break down on grounds of scarcity of resources. The breakdown of the axioms of utility theories make it possible to admit in the new paradigm a greater set of preference relations than those admitted in the standard paradigm, such as lexicographers' preferences. This makes the new paradigm more inclusive and realistic.

The new paradigm draws a distinction between engagement with a gamble and the subsequent immediate consequence of engagement with that gamble when it plays out; and it draws a distinction between the effects of these events on an individual's welfare. For an individual, engaging with a gamble, as with any other object, can generate a welfare-gain or a welfare-loss. The latter welfare effects of gambling are separate from the welfare effects of the subsequent immediate consequences of gambling represented by the *outcomes* of the gamble in play, which can give rise to *risk* for the individual (Falahati 2019b, pp. 123–24). The new paradigm thus draws a distinction between *gambling per se* and *risk-bearing per se* as a result of gambling and takes account of both. This distinction is not made under the axioms of EUT.

Therefore, the new paradigm, *unlike EUT*, takes account of the decision-maker's welfare effects of gambling per se, i.e., what Von Neumann and Morgenstern (1953, pp. 629–32) call 'utility or disutility of gambling' for the player. They refer to the enjoyment (excitement) or disdain (anxiety) that engagement with gambling can generate *separately* from the welfare effects of any risk from the outcomes of gambles, as the consequence of gambling, and go on to state (Von Neumann and Morgenstern 1953, pp. 629–30):

> 'It constitutes a *much deeper problem* to formulate a system, in which gambling has under all conditions a definite utility or disutility, where numerical utilities fulfilling the calculus of mathematical expectations cannot be defined by any process, direct or indirect. In such a system some of our axioms must be necessarily invalid. *It is difficult to foresee at this time, which axiom or group of axioms is most likely to undergo such a modification*'. (emphasis added)

EUT does not deal with this *deeper problem,* however, this is addressed in the new paradigm, where two different evaluation stages are identified which lead to determining the welfare effects of *engagement with gambles per se* initially and the *immediate consequence of this engagement* subsequently. The first stage carries with it the evaluation of the welfare effects of engagement with gambles, thereafter, the second stage starts immediately and leads to the evaluation of the outcomes of gambles, which can generate risk, as explained earlier. Each holistic decision-maker takes into account the welfare effects of both these processes, and finds their algebraic sum for himself/herself, whilst not constrained by the axioms of EUT.

***Explanations for the existence of the firm, its profit and where the extra money this profit represents comes from in a CEFE***: In the CEFE of the standard paradigm, there is no explanation for the existence of the firm, profit of the firm, or where this profit in monetary terms comes from, as the cost of inputs and the selling price of outputs produced by those inputs are assumed to be the same in terms of present values under the strong law of one price. This leads to the profit puzzle in both neoclassical economics (Desai 2008) and in classical economics (Tomasson and Bezemer 2010), as it cannot explain the existence of equity markets whilst it presumes they exist. Moreover, it cannot explain *intrinsic* economic growth, i.e., leaving aside any growth from external factors such as new technology. By contrast, the CEFE of the new paradigm, which abandons the strong law of one price, provides the following natural justification of these phenomena.

In the CEFE of the new paradigm, in a voluntary exchange of one good with another good, each party receives the welfare-gain of the good he/she acquires and gives up the welfare-gain of the good he/she disposes, with the welfare-gain of the good acquired exceeding the welfare-gain of the good disposed from each party's perspective. Thus, each party gains in terms of his/her own welfare, in the absence of which the transaction will be reversible and can be annulled. A similar phenomenon occurs when a good is exchanged with money, where for the buyer the welfare-gain of the good acquired must exceed the welfare-gain of the money paid for it, and for the seller the welfare-gain of the money received must exceed the welfare-gain of the good disposed for it. It is this subjective gain, which motivates each transaction and makes a voluntary exchange a win-win game for each party. The subjective monetary measure of this welfare-gain is available to the market participant engaged in each transaction as a buyer or a seller by assumption in the new paradigm, and it represents the required welfare-gain to compensate the market participant's cost of capital tied up during *the instant of the exchange* (Falahati 2019b, p. 121).

The latter subjective cost of capital of the market participant in each transaction must be no lower than its objective counterpart, which is expressed in terms of objective prices when these prices are observable at the date that the transaction is completed. In non-monetized transactions e.g., barters, this objective counterpart does not appear in monetary terms. In monetized transactions, this objective counterpart is visible when both the buying and selling prices of each good subject to exchange are observable from the perspective of the same trader. For example, in the case of an arbitrageur, this objective counterpart is the excess of the selling price over the buying price (at the same date) of the good subject to arbitrage. In the case of a firm, it is the excess of the selling price of its output over the buying price of its input (at the same date) which generates its output.

Therefore, for a firm as a going concern in continuous operation, its subjective monetary gain from buying its inputs and selling its outputs must be met by its objective counterpart in the form of an added present value (Falahati 2019c, p. 156), which arises from the transformation of its inputs into its outputs continually.

On the other hand, the loan input-output cycle theory of banking in the CEFE of the new paradigm (Falahati 2019c, pp. 164–65), referred to earlier, explains how endogenously in parallel with the rest of the economy, the banking system can continually create the *extra loans* and *new* money which support the generation of this added value of the firms, including banks, whilst it also can create widespread customer funding gaps.

Consequently, in the CEFE of the new paradigm, the firm emerges as the engine of generating added value from its inputs, financed by a banking system that can create the necessary credit and new money supporting it; a process the continuity of which can be halted endogenously by a banking crisis, when banks breach their debt-capacities.

In contrast, the CEFE of the standard paradigm, under the strong law of one price, denies the generation of this added value i.e., the profit of the firm as a going concern. The latter profit is also the reason for the existence of banks in the CEFE of the new paradigm, as long as they avoid generating such widespread customer funding gaps that lead to banking crises. This scenario is unlike that of the standard paradigm, which in its CEFE offers no justification for the existence of firms, banks, their profits and equity markets. It is only the new paradigm that in its CEFE presents a full explanation of the profit puzzle, existence of firms, existence of equity markets and the banking system.

## Appendix C. Application of the Mathematical Concept of Function in Utility Theory

The concept of a mathematical function plays a central role in the proof of EUT (and of ordinal utility theory). However, this crucial matter is overlooked in the existing literature. Von Neumann and Morgenstern (1953, p. 88) define a function as:

'A function $\phi$ is a dependence which states how certain entities $x, y, \ldots$-called the variables of $\phi$- determine an entity $u$ called the value of $\phi$. Thus $u$ is determined by $\phi$ and by the $x, y, \ldots$, and this determination ... will be indicated by the symbolic equation $u = \phi(x, y, \ldots)$.'

Such a function can be single-valued or multivalued or both. *The vN-M definition of a function is inconsistent with axiomatic set theory.* For, functions are single-valued (and not multivalued) in Zermelo–Fraenkel set theory, and thanks to Bernays (1888–1977) and Gödel (1906–1978), they are also single-valued (and not multivalued) in von Neumann–Bernays–Gödel set theory (Hamilton 1982, pp. 115–33, 145–55). This follows from the fact that von Neumann–Bernays–Gödel set theory is more conservative than Zermelo–Fraenkel set theory such that whatever is true in Zermelo–Fraenkel set theory is also true in von Neumann–Bernays–Gödel set theory, but not necessarily vice versa. This paper recognizes the validity of both these two separate set theories.

Let us now return to utility theory. Ordinal utility theory seeks to define formally the concept of rationality for decision-making under perfect certainty. Ordinal utility theory was developed before the era (i.e., post 1960's) when the set-theoretic definition of a function was widely employed (Kleiner 1989, p. 284), as Fishburn (1988, pp. 1–24) implicitly notes in his outline of the historical development of utility theories. Von Neumann and Morgenstern (1953) rely on ordinal utility theory to obtain the utility of each outcome of a gamble when the latter outcome is a good or money; hence they implicitly and explicitly (1953, p. 88) rely on the pre-set-theoretic notion of a function in their proof.

Propositions A1 and A2 in this appendix challenge the claims of ordinal utility theory and EUT based on set-theoretic concepts. To ensure common ground with the reader, this appendix initially runs through some basic concepts of Zermelo–Fraenkel set theory, leading to definitions of function, onto function and multifunction, and then contrasts this with the application of these concepts in ordinal utility theory and EUT, respectively. It is possible to argue that for sophisticated readers of this article, such an exposition is unnecessary. However, eminent scholars such as Varian (1992, p. 177) and Machina (2008b, p. 191) appear to be unaware that when under EUT they express utility of a gamble in terms of *indefinite* integrals, which are multifunctions, they are denying the existence of vN-M utility functions! Hence, the need for the following clarifications:

**Definitions of function, multifunction and onto function**: In examining the proofs of the existence of an ordinal utility and a vN-M utility function from a post-set-theoretic perspective, it is necessary to have the set-theoretic definitions of *function*, *multifunction* and *onto function.* That, in turn, requires the definition of a *binary relation*, which follows:

Given objects $s$ and $t$, an *ordered pair* of these objects is denoted by $(s, t)$, where the ordered pair $(s, t)$ differs from $(t, s)$ unless $s$ and $t$ are identical objects. The *Cartesian product* of two sets $S$ and $T$, denoted by $S \times T$, is the set of all ordered pairs $(s, t)$ such that $s \in S$ and $t \in T$. A *binary relation* on a set is a set of ordered pairs of the elements of that set. A binary relation from set $S$ to set $T$ is a subset of the Cartesian product $S \times T$.

A *function* from $S$ to $T$ is a binary relation $R$ from $S$ to $T$ if for each $s \in S$ there is one and only one $t \in T$ such that $(s, t) \in R$ (Hamilton 1982, p. 83; Bourbaki 1960, p. 76). This is where $S$ is the *domain* and $T$ the *codomain* of $R$, $s$ is the *input* (or *argument*) and $t$ is the *output* (or *value*) of $R$; and one can denote the function by $R : S \to T$ and write $t = R(s)$. Hence, for *every* input in the domain, there is *one and only one* output in the codomain. This is what makes a function *total* and *single-valued*.

In contrast, a *multi-valued* function or a *multifunction* (Knopp 1996, Part I, p. 103 and Part II, pp. 93–146) has for at least one input from its domain two distinct outputs in its codomain.

A function is *onto* (i.e., surjective) if its codomain is identical with the set of its outputs. A function is *bijective* if it is both onto and one-to-one, and it thus becomes *invertible* (Bartle 1976, pp. 17–21; Simon and Blume 1994, pp. 197, 365), as in Figure A1.

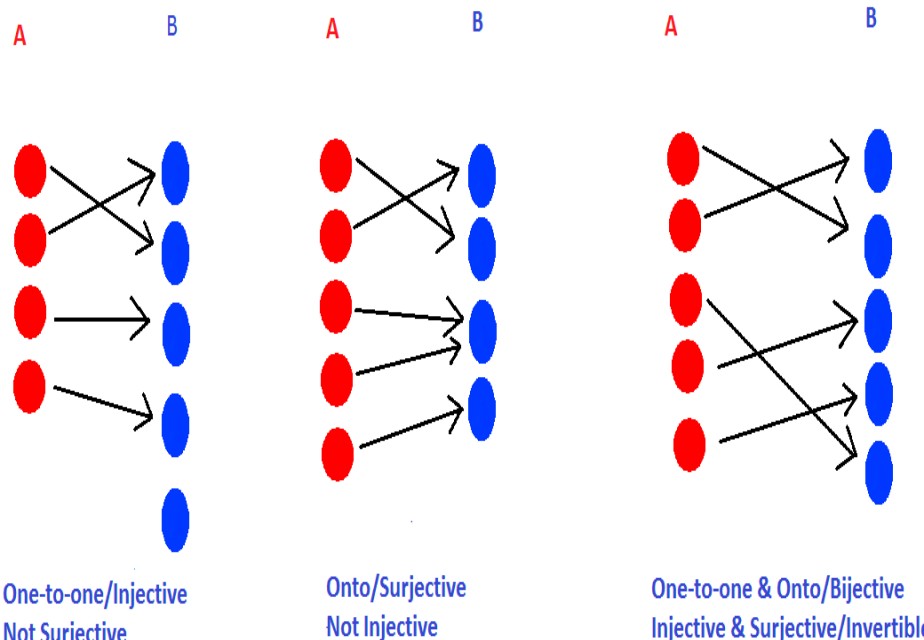

**Figure A1.** One-to-one, onto and invertible functions from set A to set B.

A multifunction from set *S* to set *T* has at least two *branches* in the form of onto functions from *S* to *T'* and from *S* to *T''*, where *T'* and *T''* are each a non-empty subset of *T*, and *T' ≠ T''*. Conversely, onto functions from *S* to *T'* and from *S* to *T''* form a multifunction from *S* to *T*, as in Figure A2.

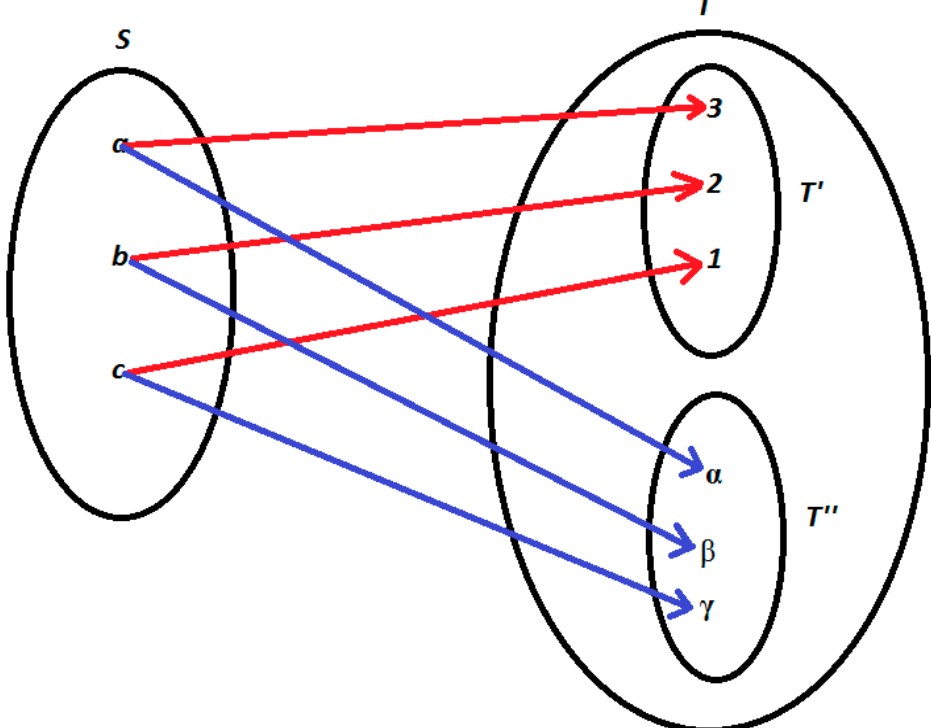

**Figure A2.** Onto functions from *S* to *T'* and from *S* to *T''* form a multifunction from *S* to *T*.

**Proposition A1.** *If axioms of ordinal utility theory on a decision-maker's preference-ordering of a set of goods hold, no binary relation from the set of goods to the set of real numbers representing this preference-ordering can exist as a function other than as an onto function which is a branch*

*of a non-unique multifunction. The existing literature fails to recognize the latter multifunctions and treats them as if they were single-valued utility functions. To avoid the emergence of these multifunctions, there must be one and only one subset of the set of real numbers representing the decision-maker's unique preference-ordering.*

**Proof.** Ordinal utility theory (Jehle and Reny 2011, pp. 3–17) claims that when a decision-maker's preference-ordering of a set of goods $S$ follows its axioms, and the set of real numbers is $T$:

(i) A binary relation from $S$ to $T$ exists such that a function $u : S \rightarrow T$ numerically represents the decision-maker's preference-ordering of the set of goods. Moreover, it deals with a monotonic transformation of $u$ as follows:

(ii) The function $v : S \rightarrow T$ will also represent the same preference-ordering as $u$, if for every $s \in S$, $v(s) = g[u(s)]$ where $g : T \rightarrow T$ is a strictly increasing function.

Let $u$ and $v$ be subsets of the binary relation $Q$ from $S$ to $T$. It follows from (ii) that for the binary relation $Q$ from each input $s \in S$, there will be two distinct outputs in $T$, namely $u(s)$ and $g[u(s)]$; thus the binary relation $Q$ is a multifunction, with $u$ and $v$ each being a branch of it, and each being an onto function, and each representing the same preference-ordering from $S$ to $T$. Hence, no binary relation from the set of goods to the set of real numbers can exist as a function representing the preference-ordering of the set of goods other than as an onto function which is a branch of a multifunction.

Clearly, any two functions with characteristics of $u$ and $v$, each of which are many, can generate a multifunction; hence such a multifunction is not unique. Therefore, the claims of ordinal utility theory lead to infinitely many onto functions. each as *a branch of a multifunction* from the set of goods to the set of real numbers, and each branch representing the same preference-ordering of the set of goods. For example, in Figure A2, if $a, b$ & $c$ were quantities of three different goods such that $a \succ b \succ c$ and $\alpha = 30, \beta = 20,$ and $\gamma = 10,$ we will have a multifunction from $S$ to $T$, with $S \rightarrow T'$ and $S \rightarrow T''$ being each an onto function and a branch of this multifunction and each representing the preference-ordering of $a, b$ & $c$.

The existing literature does not admit the existence of the foregoing non-unique multifunctions, and it assumes the axioms of ordinal utility theory always generate *only* single-valued functions. As a result, ordinal utility theory does not draw a distinction between any one of these multifunctions and its branches. In effect, supporters of *ordinal utility theory unknowingly treat the foregoing multifunctions as if they are each single-valued functions* representing the unique preference-ordering of the set of goods.

To avoid the emergence of the foregoing multifunctions, and for a real-valued ordinal utility function to exist, there must be one and only one subset of the set of real numbers $T$ representing the decision-maker's unique preference-ordering (of the set of goods $S$) which this function must represent, by the definition of a function in its set theoretic sense.

However, this characteristic is not true of the ordinal utility function. For, under ordinal utility theory, any positive monotonic transformation of an ordinal utility function must retain the preference-ordering of the set of goods as objects of choice (Jehle and Reny 2011, p. 17). This leads to infinitely many subsets of the set of real numbers $T$ (say, $T', T'', T''', \ldots$) representing the same preference-ordering, ensuring the existence of at least one multifunction from $S$ to $T$. □

**Proposition A2.** *If axioms of EUT on a decision-maker's preference-ordering of a set of gambles hold, no binary relation from the set of gambles to the set of real numbers representing this preference-ordering can exist as a function other than as an onto function which is a branch of a non-unique multifunction. The existing literature treats the latter multifunctions as if they were single-valued. To avoid the emergence of these multifunctions, there must be one and only one subset of the set of real numbers representing the decision-maker's unique preference-ordering of the set of gambles.*

**Proof.** Expected utility theory ([Jehle and Reny 2011](#), pp. 97–118) claims that when a decision-maker's preference-ordering of a set of gambles $S$ follows its axioms, and the set of real numbers is $T$, then:

(i) A binary relation from $S$ to $T$ exits such that a function $u : S \rightarrow T$ numerically represents the decision-maker's preference-ordering of the set of gambles. Moreover, it deals with a positive affine transformation of $u$ as follows:

(ii) The function $v : S \rightarrow T$ will also represent the same preference-ordering as $u$, if for every $s \in S$, $v(s) = a + b[u(s)]$, where $a$ and $b$ are real numbers with $b > 0$.

Let $u$ and $v$ be subsets of the binary relation $Q$ from $S$ to $T$. It follows from (ii) that for the binary relation $Q$ from each input $s \in S$, there will be two distinct outputs in $T$, namely $u(s)$ and $a + b[u(s)]$; thus the binary relation $Q$ is a multifunction, with $u$ and $v$ each being a branch of it and each representing the same preference-ordering from $S$ to $T$. Therefore, no binary relation from the set of gambles to the set of real numbers can exist as a function other than as an onto function which is a branch of a multifunction.

Clearly, any two functions with the characteristics of $u$ and $v$, each of which are many, can generate a multifunction; hence such a multifunction is not unique. Therefore, the claims of EUT lead to infinitely many onto functions, each as *a branch of a non-unique multifunction* from the set of gambles to the set of real numbers and each branch representing the same preference-ordering of the set of gambles. However, EUT does not draw a distinction between any one of these multifunctions and its branches. *In effect, EUT treats these multifunctions as if they were single-valued functions representing the decision-maker's unique preference-ordering of the set of gambles.*

To avoid the emergence of the foregoing multifunctions, and for a real-valued vN-M utility function to exist, there must be one and only one subset of the set of real numbers $T$ representing the decision-maker's preference-ordering (of the set of gambles $S$) which this function must represent, by the definition of a function in its set-theoretic sense. However, this characteristic is not true of the vN-M utility function. For, under EUT, any positive affine transformation of a vN-M utility function must retain the preference-ordering of the set of gambles as objects of choice ([Jehle and Reny 2011](#), p. 108). This leads to infinitely many subsets of the set of real numbers $T$ (say, $T', T'', T''', \ldots$) representing the same preference-ordering, ensuring the existence of at least one multifunction from $S$ to $T$. □

**Appendix D. Errors in Textbook Proofs of the Existence of vN-M Utility Functions**

The proofs in textbooks (e.g., [Ingersoll 1987](#), p. 10; [Varian 1992](#), pp. 171–4; [Jehle and Reny 2011](#), pp. 97–105) typically note that under the axioms of EUT, there is a worst and a best gamble for the decision-maker. This is where utility of the worst gamble $U(G_{worst})$ and utility of the best gamble $U(G_{best})$ are conveniently assigned real numbers 0 and 1, respectively, amongst the *infinitely many real numbers* which $U(G_{worst})$ and $U(G_{best})$ can each take under the vN-M definition of a function, and $U$ is claimed to be single-valued on the grounds that for every input in $U$, there will be one and only one output for it. The set of all such outputs of $U$ will be its codomain, which will make $U$ *onto* (i.e., surjective) by definition. As such, $U$ is claimed to be a vN-M utility function.

Nonetheless, a vN-M utility function $U$ must be an increasing function of wealth for those who prefer more wealth to less, and if it is strictly increasing, then it will be one-to-one (i.e., injective), whilst also being onto (i.e., surjective). Hence, if $U$ is a strictly increasing function in a valid set-theoretic sense, it must have a unique *inverse*. However, $U$ cannot be invertible, as the following theorem proves.

**Theorem.** *Any certainty equivalent derived from a strictly increasing vN-M utility function generates an internal inconsistency for this utility function.*

**Proof.** Let us take the simple case of the vN-M utility function $U$ and gamble $\underset{M+Z}{G}$ studied in Section 5.1 for when $U$ is strictly increasing, and hence one-to-one (i.e., bijective). The widely accepted textbook proof of existence of vN-M utility functions discussed earlier in this appendix points out that vN-M utility functions are onto (i.e., surjective). Hence, if $U$ exists, it must have a unique *inverse function* $U^{-1}$.

However, according to Equation (1) in Section 5.1 $U(\underset{M+Z}{G}) = U[M + E(Z) - {}_M h_Z]$, which means that there is a unique certainty equivalent $M + E(Z) - {}_M h_Z$ for gamble $\underset{M+Z}{G}$. Under EUT, for the same certainty equivalent $M + E(Z) - {}_M h_Z$, there can be infinitely many gambles such as $\underset{M+Z\prime}{G}$, where $Z\prime \neq Z$ is a random variable with another known probability distribution such that $M + E(Z\prime) - {}_M h_{Z\prime} = M + E(Z) - {}_M h_Z$ and $U(\underset{M+Z\prime}{G}) = U[M + E(Z\prime) - {}_M h_{Z\prime}]$. Hence, for the original vN-M utility function $U$ which I started with in this proof, and which is one-to-one and onto, there can be no unique inverse function $U^{-1}$, a contradiction for the existence of $U$ as a single-valued function. $\square$

**Appendix E. Errors in Proofs of von Neumann–Morgenstern, and Herstein and Milnor**

This appendix illustrates the implications of using the same notation for both the operation of *addition* and for the operation of *mixture* concurrently in Von Neumann and Morgenstern (1953) and also in Herstein and Milnor (1953). This is where a mixture can be denoted by $\oplus$, whilst the notation for addition is +. Such a distinction can distinguish the mixture $\mu a \oplus (1 - \mu)c$ from a convex combination like $\mu a + (1 - \mu)c$. This is where $a$ &$c$ are real numbers representing monetary values and $1 \geq \mu \geq 0$, as in Section 5.2. The absence of this distinction leads to contradictions and fails to satisfy the aims of these authors in proving the expected utility theorem. Indeed, it does the opposite by disproving this theorem, as the following demonstrates:

For simplicity, let us assume $X$ is a discrete random variable. Consider the case of gamble $\underset{X}{G}$ with the random variable $X$ taking monetary values equal to $X_i$ with probabilities $\lambda_i$ for $i = 1, 2, 3 \ldots n$, where $1 \geq \lambda_i \geq 0$, $\sum_{i=1}^{i=n} \lambda_i = 1$. Herstein and Milnor (1953, p. 292) take a gamble such as $\underset{X}{G}$ to be an element of mixture set $\mathfrak{M}$, where $\mathfrak{M}$ can be a *convex set in a real vector space* (Bartle 1976, p.59) and $X_i \in \mathfrak{M}$, and $\underset{X}{G} = \sum_{i=1}^{i=n} \lambda_i X_i$.

Under EUT, for a vN-M utility function with $\underset{X}{G}$ as its input to exist, there must be a real-valued function $U$ such that $U(\underset{X}{G}) = E[U(X)]$, irrespective of whether $U$ is strictly convex, which it must be for strictly risk-seekers, *or* strictly concave, which it must be for strictly risk-averters. However, taking $\underset{X}{G} = \sum_{i=1}^{i=n} \lambda_i X_i$, as in Herstein and Milnor (1953, p. 292), makes it impossible for $U(\underset{X}{G}) = E[U(X)]$ as long as $U$ is a *strictly* convex or a *strictly* concave real-valued function. For, as the literature on convexity of sets and functions (Rockafeller 1970; Bartle 1976, p. 211; Simon and Blume 1994, p. 505) makes clear, if $U$ is strictly convex, then $U(\underset{X}{G}) < E[U(X)]$, and if $U$ is strictly concave, then $U(\underset{X}{G}) > E[U(X)]$, contrary to the claims of EUT!

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
