# Peer review of "The Standard Model of Rational Risky Decision-Making"

_jrfm, doi:10.3390/jrfm14040158_

Round 1
Reviewer 1 Report
The author improved the paper based on my recommendations so I can accept it for publication.
Author Response
I thank Reviewer 1'for his/her recommendation to accept the paper for publication.
I have ensured that all spellings in UK English are correct.
As acknowledged, I have improved the paper in the light of Review 1's comments for which I am very grateful.
I have made clarifying improvements to the rest of the paper in the light of other Reviewers' views and my own clarifying thoughts.
Reviewer 2 Report
Dear Authors,
The submitted paper " The Standard Model of Rational Risky Decision-making" is addressing an important and interesting topic, therefore thank you very much for your work and the contribution. This paper claims that EUT and the axioms of utility theory, including EUT, are not consistent with scarcity of resources, hence they cannot be rational. It also explains why certain persistent observed behaviour of individuals facing risk and persistent observed behaviour in risk management in the finance industry are economically justifiable, contrary to EUT which regards such behaviour irrational.
The Authors give good justification of research conducted. Literature is relevant. I find the paper interesting and technically sound.
Generally, paper is well structured, important theoretical and practical aspects of the examined problem are studied and presented in a clear and consistent manner. Paper is well positioned in journal’s aim and scope. Thus I recommend to accept the paper.
Author Response
I am most grateful for the Reviewer 2's appreciation of the paper and kind comments.
I have ensured that all spellings in UK English are correct.
I have improved the introduction by explicitly stating the aim and the research design of the paper in lines 8-9 of the first paragraph as follows; "....this paper aims to re-examine Expected Utility Theory (EUT) by first setting out the claims of EUT and the problems that they generate, and later by identifying the generic cause of these problems and finally indicating how they can be resolved".
I have also provided a further reference regarding the notation used for a gamble in line 4 of page 3 of the introduction, where I refer to "(Varian, 1992, p. 173-176)".
I have made clarifying improvements to the rest of the paper in the light of other Reviewers' views and my own clarifying thoughts.
Reviewer 3 Report
The study is interesting and worth exploring. However, some critical points need to be addressed as stated below.
The objective of this study was not clearly presented in the introduction section. Please discuss the research questions more elaborately in the introduction.
Some of the chapters need restructuring. For example: the author can create one section Implication of the study and mention the sections 5.1. Implications of EUT for the Standard Paradigm in Economics/Finance, and 7. Broad Implications of this Paper there. Results and Discussions should be one section.
The conclusion section should be more precise in the case of the presentation of scientific contributions and limitations to the work done.
Author Response
I am grateful for the Reviewer 3's interest in, and appreciation of, the paper.
I have improved the introduction by explicitly stating the aim of the paper in lines 8-11 and setting out the research design of the paper in the first paragraph as follows: "...this paper aims to re-examine Expected Utility Theory (EUT) by first setting out the claims of EUT and the problems that they generate, and later by identifying the generic cause of these problems and finally indicating how they can be resolved".
I have also provided a further reference regarding the notation used for a gamble in line 4 of page 3 of the introduction, where I refer to "(Varian, 1992, p. 173-176)".
I have also improved the structure of the paper, for example, by dedicating Section 8 to Results and Discussion, and ensuring that the Conclusion in Section 10 is accurate, succinct and supported fully by the Results. The Conclusion In Section 10 also brings to readers' attention the limitations in describing the new paradigm, and the potential of the new paradigm for resolving other puzzles generated by the standard paradigm in this article by stating " To help this new paradigm being more widely recognized, there is a need to elucidate it more, and to expand it by demonstrating its potential for resolution of more anomalies in the future."
I have made clarifying improvements to the rest of the paper in the light of other Reviewers' views and my own clarifying thoughts.
This manuscript is a resubmission of an earlier submission. The following is a list of the peer review reports and author responses from that submission.
Round 1
Reviewer 1 Report
see pdf.

Reviewer 2 Report
The paper analyses a real problem in the field of risk management of financial decisions. The article represents a very high scientific level, especially due to its comparative approaches in connection with the paradigms and the very professional methodology.
However, the structure of the paper is not appropriate, in its present form the real values and the main results are not easy to identify and not understandable for the readers enough.
Before publishing, I recommend some minor improvements, mainly from a restructuring point of view.
1) The abstract should be more comprehensive.
2) There is no literature review chapter in the paper; some sources are processed, analyzed, and cited in the Introduction part, those could be a good base for a separate comparative and critical literature review.
3) The methodology is not described properly, a separate chapter (Material and methods or Methodology) should be inserted, and the related text restructured in this way.
4) I don't like the idea to have discussions and conclusions in one chapter, it should be separated and the previous chapters should be integrated under a new "Results and discussions" chapter
5) the limitations of the research should be indicated in a more clearly way.